# β-Ionone: Its Occurrence and Biological Function and Metabolic Engineering

**DOI:** 10.3390/plants10040754

**Published:** 2021-04-12

**Authors:** Antonello Paparella, Liora Shaltiel-Harpaza, Mwafaq Ibdah

**Affiliations:** 1Faculty of Bioscience and Technology for Food, Agriculture, and Environment, University of Teramo, Via Balzarini 1, 64100 Teramo, Italy; apaparella@unite.it; 2Migal Galilee Research Institute, P.O. Box 831, Kiryat Shmona 11016, Israel; lioraamit@bezeqint.net; 3Tel Hai College, P.O. Box 831, Kiryat Shmona, Upper Galilee 12210, Israel; 4Newe Yaar Research Center, Agricultural Research Organization, P.O. Box 1021, Ramat Yishay 30095, Israel

**Keywords:** biological activity, biosynthesis, β-ionone, insect attractant/repellant, metabolic engineering

## Abstract

β-Ionone is a natural plant volatile compound, and it is the 9,10 and 9′,10′ cleavage product of β-carotene by the carotenoid cleavage dioxygenase. β-Ionone is widely distributed in flowers, fruits, and vegetables. β-Ionone and other apocarotenoids comprise flavors, aromas, pigments, growth regulators, and defense compounds; serve as ecological cues; have roles as insect attractants or repellants, and have antibacterial and fungicidal properties. In recent years, β-ionone has also received increased attention from the biomedical community for its potential as an anticancer treatment and for other human health benefits. However, β-ionone is typically produced at relatively low levels in plants. Thus, expressing plant biosynthetic pathway genes in microbial hosts and engineering the metabolic pathway/host to increase metabolite production is an appealing alternative. In the present review, we discuss β-ionone occurrence, the biological activities of β-ionone, emphasizing insect attractant/repellant activities, and the current strategies and achievements used to reconstruct enzyme pathways in microorganisms in an effort to to attain higher amounts of the desired β-ionone.

## 1. Introduction

Plant volatile organic compounds (VOCs) are specialized metabolites, which have been described from more than 90 plant families [1,2]. A wide diversity of compounds are emitted from leaves, flowers, fruits, bark, and roots, as well as from specialized storage structures, e.g., glands [1,2]. VOCs are lipophilic liquids and have low molecular mass (<300 Da) with high vapor pressures. Plant VOCs are derived from various biosynthesis pathways and are often classified into significant groups, including phenylpropanoids, fatty acids, amino acids, and terpenoids [3]. 

VOCs are primarily intended to protect plants against pathogens and herbivores or to provide a reproductive benefit in attracting pollinators and seed dispersers [3,4,5,6,7]. Plant VOCs have attracted considerable scientific interest for commercial applications, e.g., decorative cosmetics, shampoos, and fine fragrances, as well as for their potential benefits for human health [8,9,10]. 

Ionone is a ketone with a monocyclic terpenoid backbone made up of 13 carbon atoms. The name “ionone” is derived from “iona” (Greek for violet), which refers to the violet odor, and “ketone,” which refers to the structure of the ionone [11,12]. β-Ionone is the 9,10 and 9′,10′ cleavage product of β-carotene (Figure 1) [13,14,15,16]. 

It is a common aromatic volatile found in various tissues in many plant species, e.g., in the essential oil (EO) of *Osmanthus fragrans, Petunia hybrid,* and *Rosa bourboniana*, and in fruits and vegetables, including carrot, tomato, melon, raspberry, apricot, plum, apple, and fig [14,17,18,19,20,21]. β-Ionone has become quite attractive for commercial applications and in medical and pharmaceutical industries [22,23,24,25,26,27,28]. 

This review aims to highlight the latest advances concerning the occurrence and the biological activities of β-ionone, and the current state of knowledge about its application in different industrial environments. 

## 2. Occurrence of β-Ionone

Headspace solid-phase microextraction (HS-SPME) was used to obtain the EO of *Osmanthus fragrans*, which was then analyzed by gas chromatography–mass spectrometry (GC–MS) at four different stages of flowering. The primary chemical components of the EO obtained from *O. fragrans* were β-ionone, *α*-ionone, and other VOCs, e.g., linalool and nerol [29]. Capillary GC–MS with olfactometry analysis was used to determine the volatile profile of samples of *Thymus vulgaris* EO. The compounds with the highest aroma impact for *T. vulgaris* were β-ionone, borneol, terpinyl acetate, and β-damascenone [30]. The EO of the leaves of *Lawsonia inermis* L. (henna) was found to contain β-ionone as one of its major components [31]. The EO of the genus *Rosa* (Rosaceae) is commercially crucial in the fragrance and food industries [32,33,34]. Honarvar et al. [35] reported that EOs of fresh and dried flowers of Musk Rose contain approximately 2% of β-ionone of the total chemical constitutes. In the EO obtained from fresh aerial parts of *Viola tricolor* (Violaceae), Anca et al. [36] identified 35 VOCs, and β-ionone constituted around 2% of the total. The chemical composition of the EO from different stages of *Medicago marina* was analyzed, and β-ionone was one of the main constituents of the flowering samples (6%) but increased during the vegetative stage (14%) [37]. β-Ionone was also determined at different percentages in the EOs of Moroccan herbs, namely *Myrtus communis*, *Cistus ladanifer*, and *Montpellier cistus* [38]. In contrast, in the EO of maca (*Lepidium meyenii*), β-ionone was found to be a minor component [39] (Figure 2). 

The aroma of fruits and vegetables is determined by unique combinations of VOCs, including β-ionone (Figure 3), although different fruits, roots, and vegetables often share many aroma characteristics. Yahyaa et al. [15] analyzed the VOC composition in the roots of various carrot genotypes with different colors (orange, purple, white, and yellow). They found that only orange and purple carrot roots accumulate β-ionone; white and yellow carrot roots do not [15]. β-Ionone was also identified in the chemical compositions of EOs of *Daucus carota* subsp. *sativus* [40]. Guillot et al. [41] characterized the aromatic profile of several apricot cultivars (Iranien, Orangered, Goldrich, Hargrand, Rouge du Roussillon, and A4025) by using HS-SPME. In the apricot, several VOCs, including β-ionone, were recognized by HS-SPME–GC–Olfactometry methods as responsible for its aromatic notes [41,42,43,44]. Tomato fruit contains a complex mixture of VOCs and non-volatile compounds that contribute to the fruit’s overall aroma and taste, and β-ionone is involved in tomato’s flavor [45]. Similarly, β-ionone was found to be a significant VOC in melon and watermelon [14,45]. Among stone fruits, in the plum (*Prunus domestica*), SPME–GC–MS analysis of the VOCs revealed that β-ionone is one of the characteristic odor active compounds in fresh plums, with concentrations far above the odor threshold [20,46]. The carotenoid-derived volatile β-ionone plays an essential role in forming green and black tea flavors due to its low odor threshold [47]. Moreover, this volatile, together with other VOCs, e.g., linalool and eugenol, contributes to the “spicy” character of hops (*Humulus lupulus* L.), which is considered to be a desirable attribute in beer, associated with “noble” hop aroma [48] (Figure 3).

β-Ionone is an essential contributor to fragrance in the flowers of *Boronia megastigma* [49], in petunia (*Petunia hybrida*) [50], and in the yellow and red *Lawsonia inermis* L. (henna) flowers [51] (Figure 4). Sweet osmanthus (*Osmanthus fragrans*) is one of China’s ten most well-known flowers, and it is recognized as both an aromatic plant and an ornamental flower, with β-ionone as a major aromatic component [29,52,53]. The flowers of the mangrove plant, *Nypa fruticans*, are pollinated by various pollinators, including birds, bats, and insects. The floral scent chemistry of *N. fruticans* is composed mainly of carotenoid derivates: β-ionone, *α*-ionone, dihydro-β-ionol, and dihydro-β-ionone [54]. These floral scents might be a critical factor in attracting pollinators. *Acacia farnesiana* flowers’ volatile components contain β-ionone as one of the main components [55]. β-Ionone is also the main volatile in the *Thevetia peruviana* flowers [56].

## 3. Biosynthesis, Extraction, or Formation of β-Ionone during Processing and Storage of Foods

### 3.1. Biosynthesis of β-Ionone

The specific oxidative cleavage of carotenoids at (9,10 and 9′,10′) leads to the production of β-ionone and is catalyzed by a carotenoid cleavage dioxygenase (CCD1) (Figure 1) [13,14,15]. The biosynthesis of β-ionone in general in the plant kingdom has been recently intensively reviewed by Shi et al. [57] and Aloum et al. [58].

The CCD family is ancient and well-represented in animals, bacteria, and plants. Members of the CCD family share the following characteristics: (i) they require Fe^2+^ for catalytic activity; (ii) they have four conserved histidines that are thought to coordinate iron binding, and (iii) they have a conserved peptide sequence at their carboxyl terminus that is considered to be a signature sequence of the family [13]. CCDs frequently exhibit a high degree of region specificity for the position of the double of their substrates [12,13]. Plant CCDs can be classified into six subfamilies according to the cleavage position and/or their substrate preferences: CCD1, CCD2, CCD4, CCD7, CCD8, and 9-cis-epoxy-carotenoid dioxygenases (NCEDs) [13]. Plant CCD1 and CCD4 are primarily involved in the biosynthesis of apocarotenoids, which play an important role in herbivore–plant communication and contribute to the flavor and aroma of fruits and flowers [59,60]. CCD1 genes and enzymes involved in β-ionone formation have been identified and characterized in carrot roots, tomatoes, melons, strawberries, *Laurus nobilis*, Rose damascene Petunia flowers, and other plant species [14,15,16,17,33,50]. Characterization of CCD4 enzymes from several plant species demonstrated 9,10 (9′,10′) double bond cleavage activity to yield β-ionone formation [61,62].

### 3.2. Extraction or Formation of β-Ionone during Processing and Storage of Foods

Formation of β-ionone is a common feature in foods that are rich in β-carotene. Pénicaud et al. [63] reviewed the mechanisms of degradation of β-carotene in different fruits and vegetables, highlighting the kinetics during processing and storage. In particular, these authors proposed to apply a modeling approach, in line with what has already been done with the Maillard reaction in foods. In this way, the degradation of β-carotene, which is particularly important in the juice industry, could be controlled and reduced.

In a fruit model system, Limbo et al. [64] found that the production of β-ionone corresponded to changes in the redness/greenness index (a***), whereas the degradation of β-carotene was indicated by modification of the lightness index (L***). Degradation of β-carotene and formation of β-ionone can depend on the protective effect given by the food structure. This phenomenon has been particularly studied in heat-treated tomato products [65], where heat treatment modifies the structure of cell walls and organelles, and β-carotene with its two large β-ionone rings can scarcely reorganize.

β-Ionone is also important in the tea industry because it can be responsible for woody and floral odor in tea. In this food product, the enzymatic oxidation of catechins causes the oxidation of β-carotene and, therefore, the formation of β-ionone. Gong et al. [66] compared electronic nose and sensory analysis to identify the aroma profile of three Longjing tea samples and found that β-ionone was one of the VOCs with the highest Odor Activity Value.

In raspberry, β-ionone was found to be one of the major VOCs [67]. Hansen et al. [68] studied the extraction of *α*-ionone and β-ionone from raspberry and found that sodium chloride had a negative effect on the extraction of these compounds. The same authors demonstrated that the detection and characterization of ionones, carried out by a headspace SPME–chiral-GC–MS method, can be used to assess the authenticity of raspberry flavor in foods.

In general, when the sensory profile of fruit and fruit-based products needs to be determined, β-ionone is considered an important flavor contributor. In this respect, great care has to be used in the choice of panelists. In fact, Plotto et al. [69] found that 50% of the panelists who participated in the study on deodorized orange juice could not perceive β-ionone, while they were able to perceive *α*-ionone. This specific anosmia for β-ionone is a possibility that should be considered in the sensory analysis of fruit-based foods.

## 4. Biological Activities of β-Ionone

### 4.1. Role of β-Ionone in Plants

Plant apocarotenoids, including β-ionone, comprise flavors, aromas, pigments, growth regulators, and defense compounds. They serve as ecological cues, have roles as insect attractants or repellants, and possess antibacterial and fungicidal properties [57,58,70]. β-Ionone and other apocarotenoid pigments are also economically important because they show extremely potent odor thresholds, at a level of 0.007 ppb. Therefore, apocarotenoids, including β-ionone, although generally present at relatively low levels in fruits, often exert strong effects on the aroma’s overall human appreciation [11,57,58,71]. The synthesis and accumulation of β-ionone and its derivatives can also occur when plants are exposed to photooxidative stress through reactive oxygen species (ROS) oxidation of β-carotene [2,57,58].

### 4.2. β-Ionone as an Insect Attractant/Repellant Volatile Compound

β-Ionone is a VOC that plants produce to protect themselves in their interactions with other organisms and the environment. β-Ionone in plants appears mainly involved in defense mechanisms [72,73].

The biological effects of β-ionone have been reported for different insects [74,75,76]. It has been shown that β-ionone released from the leaves of *Trifolium glanduliferum* and *Trifolium strictum* acts as a feeding deterrent to red-legged earth mites (*Halotydeus destructor*) [77] (Figure 5). β-Ionone emitted from the flowers of *O. fragrans* repels the cabbage butterfly (*Pieris rapae*) [78] and also inhibits the movement of the crucifer flea beetle (*Phyllotreta cruciferae*) [79]. Moreover, it has been shown that males of orchid bees, *Euglossa mandibularis* (Hymenoptera, Apidae), were attracted to the β-ionone scent traps [74,80]. β-Ionone is part of an induced defense in canola, *Brassica napus*, and is released following herbivores’ wounding. By measuring the oviposition deterrence of Silverleaf whitefly (*Bemisia tabaci* Gennadius), [76] determined that β-ionone used at 0.05, 0.1, and 0.5 ng/μL acted as a deterrent (Figure 5).

Previously, Wei et al. [75] determined that β-ionone acts as a feeding deterrent for flea beetles (orchid bees, *Euglossa mandibularis*, Hymenoptera, Apidae Goeze). The authors of [79], using a T-shaped olfactometer to test the crucifer flea beetle’s behavior with a series of volatiles found in canola *Brassica napus*, observed the most potent response with β-ionone compared to other VOCs, e.g., (±)-linalool, (−)-β-pinene, *E-*β-caryophyllene, and (+)-limonene (Figure 6).

Dose responses for several compounds extracted from *Trifolum glanduliferum* were determined in spider mite *Trifolum urticae* bioassays. Of these, β-ionone, α-ionone, and coumarin showed adequate deterrence at the lowest concentrations (0.01%) [81]. In a tick-climbing repellency bioassay, β-ionone found in the EO of *Gynandropsis gynandra* exhibited one of the most repellent responses against the livestock tick *Rhipicephalus appendiculatus* [82]. In contrast, β-ionone as a constituent of mulberry leaf tea exhibited the highest attractant activity to *Lasioderma serricorne* F. (Coleoptera: Anobiidae), with a 64.4% response index at 0.1 mg/vial, and tended to have a high attractant level at decreasing doses down to 1.0 μg/vial [83]. Höckelmann et al. [84] studied the chemotaxis response of the freshwater nematode *Bursilla monhystera* (Rhabditidae) and the terrestrial model organism *Caenorhabditis elegans* (Rhabditidae) to odors produced by cyanobacteria. Electroantennogram and behavioral responses of *Cotesia plutellae* to cruciferous vegetable volatiles showed that β-ionone significantly attracts the parasitoid wasp *C. plutellae* [85].

### 4.3. Potential of β-Ionone in Human Health

The potential of β-ionone and related compounds has been recently reviewed by Ansari et al. [23]. Moreover, different studies have documented the specific properties and potential benefits of β-ionone in human health regarding important pharmacological activities: anti-inflammatory, cancer-preventing, antibacterial, antifungal, and antileishmanial [23,24,25,26,27,28].

The data from the Asokkumar et al. [86] study suggest that dietary β-ionone possesses a potent ability to suppress benzo(a)pyrene-induced lung carcinogenesis, and this suppression is at least partly attributed to the improvement of antioxidant enzymes and inhibition of cell proliferation. Antitumor activities of β-ionone have been demonstrated in melanoma, breast cancer, and chemical-induced rat carcinogenesis [87]. Recently, Xie et al. [88] have demonstrated that the prostate-specific G protein-coupled receptor activation with its ligand β-ionone can suppress prostate cancer cell growth in both in vitro and in vivo models.

Very recently, Aloum et al. (2020) proposed different mechanisms or signaling cascades to explain the relationship between β-ionone’s structure and its postulated health effects. However, based on the current literature, these authors concluded that the relationship between structure and activity is still unresolved.

## 5. Microbial Production and Biotransformation of β-Ionone

Currently, the production of β-ionone is mainly dependent on chemical synthesis. The extraction of β-ionone from various plant species is limited, as it is for several other plant natural compounds, by low recovery rates and the complexity of the mixtures obtained from plants. Furthermore, the total chemical synthesis of β-ionone is economically impractical. As a result, one appealing alternative is to express the specific plant biosynthetic pathway genes in microbial hosts such as *Saccharomyces cerevisiae* or *Escherichia coli*, and then engineer the metabolic pathway/host to improve metabolite production [89,90]. The expression of heterologous genes using high-copy-number plasmids is a widely used approach for achieving high titers of secondary metabolites in biofactories. In most cases, however, this technique is unsuccessful because it necessitates selective media use, causes genomic instability, and creates a high metabolic burden on the cell [91]. Genomic integration of expression cassettes, on the other hand, is a more convenient strategy because the resulting recombinant strains are generally more stable, and their gene expression is more controllable [92,93]. However, the small number of selection markers, integration sites, and genetic arrangements impedes strain design, as the acceptable gene dosage (copy number) for construction is restricted. CRISPR/Cas9 technology advancement in lees minimizes the need for selection markers, particularly when the transformation of *S. cerevisiae* has proven to be highly successful [94,95].

Various high-value plant natural compounds, such as pharmaceutical compounds (e.g., artemisinin), fragrance (e.g., nootkatone), biofuels (e.g., ethanol and isobutanol), and food chemicals (e.g., vanillin and resveratrol), were produced as end products in microbial systems [96]. Because of the rapid growth of microorganisms compared to plants, the accessibility of genetic manipulation, and the well-established metabolic engineering tools developed to be used in the microbial system, microbial production offers many advantages over each field and plant cell cultivation. This system can also produce much purer desired product from cell cultures than is obtainable from plant tissues or plant cultures, allowing for more straightforward (and “greener”) purification strategies. The functional reconstruction of specific plant biosynthetic pathway genes and enzymes in the microbial system, as well as the application of microbial biosynthesis for the industrial production of essential compounds, remain difficult tasks [96,97,98,99].

In addition to a comprehensive understanding of the biochemical pathway of interest, the microbial industrial production of the compound aims to provide information about the interaction between the pathway members (for example, substrate channeling, protein–protein interactions, side reactions), as well as precursor availability, high growth, production capacity, etc. Because of the extended plant growth cycle and low concentration of β-ionone, its direct extraction from plants affords the terpenoid’s low yields, insufficient to meet demand [100,101]. In fact, the annual industrial production of β-ionone is approximately 4000–8000 tonnes, and the demand is rapidly increasing [102]. In particular, β-ionone and its derivatives have attracted great interest for their commercial application in decorative cosmetics, shampoos, fine fragrances, and toilet soaps [22].

One of the first platforms used for β-ionone production was *S. cerevisiae*, reaching titers in the range of 0.2 to 5 mg/L [102,103]. Recently, Werner et al. [104], in two-phase fermentation in shake flasks, reported 180 mg/L of β-ionone production using a β-carotene hyper-producer strain. Nevertheless, this specific yeast strain has poor performance under industrial fermentation conditions [105]. β-Ionone production in *E. coli* has also been attempted. Zhang et al. [106] reported titers of 32 mg/L β-ionone in shake flasks and 500 mg/L in bioreactors. Instead, using the oleaginous yeast *Yarrowia lipolytica*, the authors of [90] obtained titers of 60 mg/L in shake flasks and 380 mg/L of β-ionone in a fermenter. More recently, López et al. [95] developed several recombinant *S. cerevisiae* strains capable of producing increasing amounts of β-ionone (33 mg/L) and its precursor, β-carotene (33 mg/L), in the culture medium after 72 h cultivation in shake flasks.

## 6. Conclusions

In conclusion, promising benefits for agriculture and humans have been reported for the apocarotenoid β-ionone, extracted from various plant species. Although several studies proved that β-ionone possesses biological activity as an insect attractant/repellant volatile organic compound, further studies are recommended to investigate the biological activity aspect of insects. Recent studies point out advances in the genetically engineered biosynthesis of β-ionone. However, much still needs to be done to develop a performing heterologous system for an economically feasible industrial production process.

## Figures and Tables

**Figure 1 plants-10-00754-f001:**
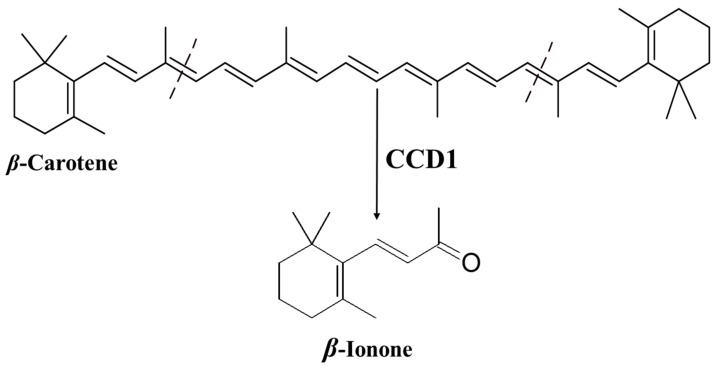
Production of β-ionone from β-carotene by carotenoid cleavage dioxygenase 1 (CCD1).

**Figure 2 plants-10-00754-f002:**
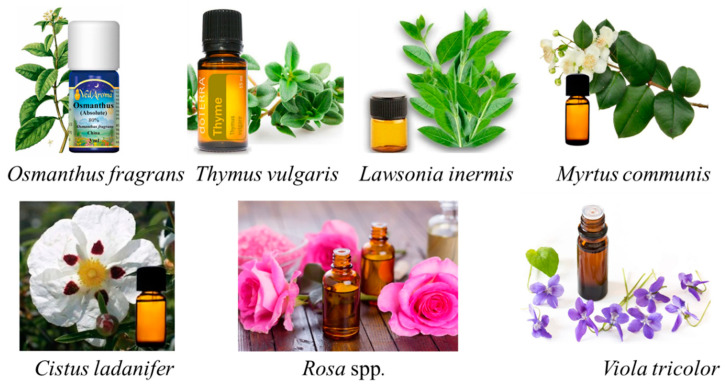
β-Ionone as a fragrance compound in the essential oils of various plant species.

**Figure 3 plants-10-00754-f003:**
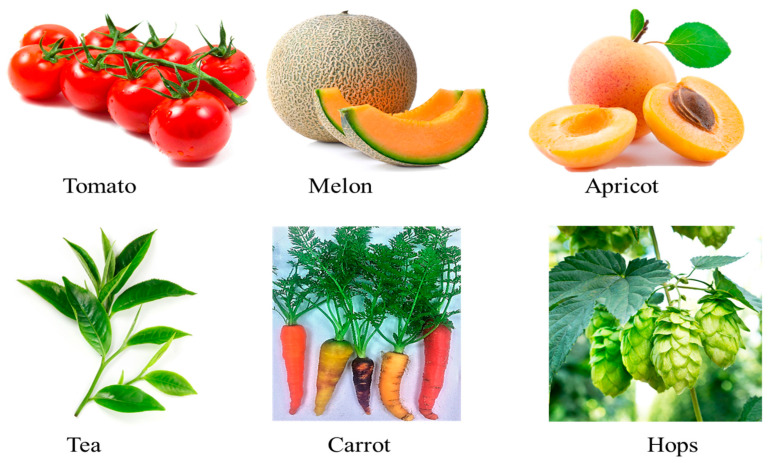
β-Ionone as an aroma compound found in some representative fruits and vegetables.

**Figure 4 plants-10-00754-f004:**
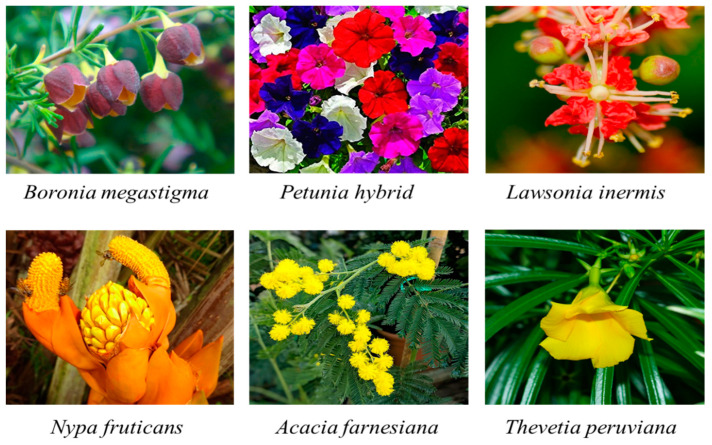
Representative flowers that accumulate β-ionone as a fragrance compound.

**Figure 5 plants-10-00754-f005:**
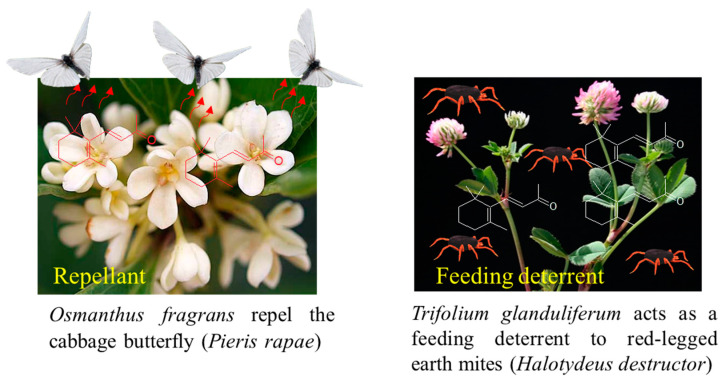
β-Ionone as an insect repellent and/or feeding deterrent found in some plant species.

**Figure 6 plants-10-00754-f006:**
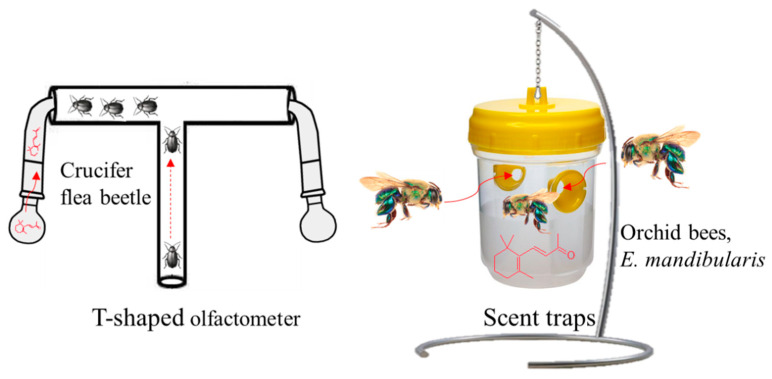
T-shaped olfactometer and scent traps using β-ionone as an insect attractant for crucifer flea beetle and orchid bee, *Euglossa mandibularis*.

## Data Availability

Not applicable.

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
