# Peer review of "β-Ionone: Its Occurrence and Biological Function and Metabolic Engineering"

_plants, 2021, doi:10.3390/plants10040754_

Round 1

Reviewer 1 Report

This review focuses on one natural compound ß-ionone that is widespread in plants. Also, this substance is currently considered as a promising anti-cancer and anti-inflammatory drug candidate. More recently (May and December 2020), the reviews on this topic were published in the Molecules (DOI: 10.3390/molecules25245822) and Journal of Chemistry (DOI: 10.1155/2020/2526956). Nevertheless, the authors hoped to bring something new to this area. According to them, the purpose of the review is "β-ionone occurrence, the biological activities of β-ionone emphasizing the insect attractant/repellant activities, and the current strategies and achievements used to reconstruct enzymes pathway in microorganisms".

I find this review to be very weak, but difficult to read. The same sentences are repeated several times, albeit in a slightly modified form. For example, Line 49 "β-Ionone is a carotenoid-derived C13 terpenoid" and Line 50-51 "β-Ionone is the 9,10 and 9', 10' cleavage product of β-carotene".

Some sections are described very superficially, for example, biological activity or biosynthesis (for some reason, instead of the word biosynthesis, authors used the "mechanism of formation", which is completely illiterate). 

As a result, no new conclusions other than "promising benefits for agriculture and humans have been reported for the apocarotenoid β-ionone" were made. 

Thus, I do not consider this review to be a significant contributor to this field. However, it may become more publishable if authors describe in detail its novelty and significance, and actually review the current literature on key issues.

Author Response

Reviewer 1

This review focuses on one natural compound ß-ionone that is widespread in plants. Also, this substance is currently considered as a promising anti-cancer and anti-inflammatory drug candidate. More recently (May and December 2020), the reviews on this topic were published in the Molecules (DOI: 10.3390/molecules25245822) and Journal of Chemistry (DOI: 10.1155/2020/2526956). Nevertheless, the authors hoped to bring something new to this area. According to them, the purpose of the review is "β-ionone occurrence, the biological activities of β-ionone emphasizing the insect attractant/repellant activities, and the current strategies and achievements used to reconstruct enzymes pathway in microorganisms".

Response: in general, we aim to write a mini-review, emphasizing the biological activities of β-ionone as an insect attractant/repellant compound. Also, we discussed and included the recent literature on β-ionone, e.g. from Shi et al., 2020 and Aloum et al., 2020. With all respect to the literature on β-ionone, including the recent one from Shi et al., 2020 did not emphasize the biological activities of β-ionone as an insect attractant/repellant compound.

I find this review to be very weak, but difficult to read. The same sentences are repeated several times, albeit in a slightly modified form. For example, Line 49 "β-Ionone is a carotenoid-derived C13 terpenoid" and Line 50-51 "β-Ionone is the 9,10 and 9', 10' cleavage product of β-carotene".

Response: again, we aim to write a mini-review and not a comprehensive review by emphasizing the biological activities of β-ionone as an insect attractant/repellant compound.

We have also changed the sentence to read as follows:  Ionone is a ketone with a monocyclic terpenoid backbone made up of 13 carbon atoms. The name "ionone" is derived from "iona" (Greek for violet), which refers to the violet odor, and "ketone," which refers to the structure of the ionone.

Some sections are described very superficially, for example, biological activity, or biosynthesis

Response: The biosynthesis of β-ionone and the genes and enzymes involving in β-ionone formation has been previously in detail reviewed. Therefore, the mini-review aims to emphasize the biological activities of β-ionone as an insect attractant/repellant volatile organic compound.

(for some reason, instead of the word biosynthesis, authors used the "mechanism of formation", which is completely illiterate). 

Response: corrected.     

As a result, no new conclusions other than "promising benefits for agriculture and humans have been reported for the apocarotenoid β-ionone" were made. 

Response: we expand the conclusion section by including the following sentences: “Although, several studies proved that β-ionone possesses a biological activity as an insect attractant/repellant volatile organic compound. However, further studies are recommended to investigate the biological activity aspect of insects.”

Thus, I do not consider this review to be a significant contributor to this field. However, it may become more publishable if authors describe in detail its novelty and significance, and actually review the current literature on key issues.

Response: we discussed and included the recent literature on β-ionone, e.g. from Shi et al., 2020 and Aloum et al., 2020

Reviewer 2 Report

This article reviews different aspects of β-ionone. In general, the article is interesting, although it could have gone into more depth in some aspects, especially in the enzymology of the ionone and in the bioengineering for its production.

Some observations:

-I would suggest joining section 2 “Mechanism of Ionone Formation” with section 4 “Extraction or formation of ionone during processing and storage of foods”.

-I would also suggest to renumber subsection 5.4 as an independent section, since it is not related to the others subsections in section 5.

Line 20: Change fruit by fruits.

Line 68: Bioavailability has not been addressed  along the paper.

Line 306: The sentence “Due to the long………….to meet the demand [93,94]” seems out of context. Probably the whole paragraph should be reorder.

Author Response

Reviewer 2

This article reviews different aspects of β-ionone. In general, the article is interesting, although it could have gone into more depth in some aspects, especially in the enzymology of the ionone and in the bioengineering for its production.

Response: again, the biosynthesis of β-ionone and the genes and enzymes involving in β-ionone formation have been previously reviewed in detail. However, we expand the section on the biosynthesis and the bioengineering of β-ionone, and we include the recent literature.

Some observations:

-I would suggest joining section 2 “Mechanism of Ionone Formation” with section 4 “Extraction or formation of ionone during processing and storage of foods”.

Response: we agree with the reviewer and combined the two sections, now under section 3.

-I would also suggest to renumber subsection 5.4 as an independent section, since it is not related to the others subsections in section 5.

Response: corrected.

Line 20: Change fruit by fruits.

Response: done.

Line 68: Bioavailability has not been addressed along the paper.

Response: we agree with the reviewer, and we deleted the bioavailability term. 

Line 306: The sentence “Due to the long………….to meet the demand [93,94]” seems out of context. Probably the whole paragraph should be reorder.

Response: corrected.

Round 2

Reviewer 1 Report

The authors have made changes to the manuscript. And although I continue to believe that the scientific value of such a mini-review is not very great, the manuscript can be published if the Editor deems it possible. 

Reviewer 2 Report

The paper has been improved and in my opinion now is suitable for publication